# Associations of Serum 25(OH)D, PTH, and β-CTX Levels with All-Cause Mortality in Chinese Community-Dwelling Centenarians

**DOI:** 10.3390/nu15010094

**Published:** 2022-12-24

**Authors:** Bin Wang, Xiaowei Cheng, Shihui Fu, Ding Sun, Weiguang Zhang, Weicen Liu, Xinyu Miao, Qing Luo, Hao Li, Jie Zhang, Xinye Jin, Yali Zhao, Yao Yao, Yizhi Chen

**Affiliations:** 1Department of Nephrology, Hainan Hospital of Chinese PLA General Hospital, Hainan Province Academician Team Innovation Center, Sanya 572013, China; 2Senior Department of Nephrology, The First Medical Center of Chinese PLA General Hospital, Chinese PLA Institute of Nephrology, State Key Laboratory of Kidney Diseases, National Clinical Research Center for Kidney Diseases, Beijing Key Laboratory of Kidney Diseases, Beijing 100853, China; 3Department of Cardiology, Hainan Hospital of Chinese PLA General Hospital, Sanya 572013, China; 4Department of Endocrinology, The Second Medical Center & National Clinical Research Center for Geriatric Diseases, Chinese PLA General Hospital, Beijing 100853, China; 5Central Laboratory, Hainan Hospital of Chinese PLA General Hospital, Sanya 572013, China; 6Center for Healthy Aging and Development Studies, National School of Development and Raissun Institute for Advanced Studies, Peking University, Beijing 100871, China; 7Center for the Study of Aging and Human Development and Geriatrics Division, Medical School of Duke University, Durham, NC 27708, USA

**Keywords:** centenarians, all-cause mortality, β-*C*-terminal telopeptide of type 1 collagen, parathyroid hormone, 25-hydroxyvitamin D

## Abstract

This longitudinal cohort study explored the associations of 25-hydroxyvitamin D [25(OH)D], parathyroid hormone (PTH), and β-*C*-terminal telopeptide of type 1 collagen (β-CTX) levels with all-cause mortality in centenarians. The study included 952 centenarians (81.4% female). During a median follow-up of 32 months, 752 (78.9%) centenarians died. The estimated 1-year, 3-year, and 5-year survival rates were 80.0%, 45.7%, and 23.6%, respectively. The association of mortality with 25(OH)D was linear, whereas the associations with PTH and β-CTX were J-shaped, with a lower risk below the median levels. Compared with 25(OH)D of ≥30 ng/mL, 25(OH)D < 30 ng/mL was associated with increased mortality (HR 1.52, 95% CI 1.24–1.86, *p* < 0.001). Compared with PTH of ≤65 pg/mL, PTH > 65 pg/mL was associated with increased mortality (HR 1.30, 95% CI 1.08–1.56, *p* = 0.005). Compared with β-CTX of <0.55 ng/mL, β-CTX ≥ 0.55 ng/mL was associated with increased mortality (HR 1.30, 95% CI 1.10–1.54, *p* = 0.002). A higher β-CTX level (even in the clinical reference range of 0.55–1.01 ng/mL) was associated with increased mortality (HR 1.23, 95% CI 1.04–1.47, *p* = 0.018). Centenarians with 25(OH)D < 30 ng/mL, PTH > 65 pg/mL, and β-CTX ≥ 0.55 ng/mL had a 2.77-fold (95% CI 1.99–3.85, *p* < 0.001) increased risk of mortality when compared with those with 25(OH)D of >30 ng/mL, PTH < 65 pg/mL, and β-CTX < 0.55 ng/mL. Lower serum 25(OH)D and higher PTH and β-CTX were independently correlated with increased all-cause mortality in Chinese community-dwelling centenarians.

## 1. Introduction

One hundred years is considered to be close to the maximum human lifespan, and centenarians provide an ideal model for investigating traits of longevity among individuals in whom age-related diseases have been largely avoided or delayed [1]. Remodeling of bone, including resorption and formation, is a continuous process throughout life and continues even in extreme senescence. Bone remodeling is affected by 25-hydroxyvitamin D [25(OH)D] and parathyroid hormone (PTH) levels and is reflected by levels of bone turnover markers. Procollagen type 1 *N*-terminal propeptide (P1NP) and β-*C*-terminal telopeptide of type 1 collagen (β-CTX) are specific markers for bone formation and resorption, respectively [2,3].

Vitamin D has been shown to have a protective effect on cardiovascular risk in both observational studies and randomized clinical trials [4]. Vitamin D might have favorable interactions with the cardiovascular system, including inhibition of the renin–angiotensin-aldosterone pathway, insulin sensitization, immune regulation, and antifibrotic, antihypertrophic, and antiatherosclerotic effects [5]. Vitamin D deficiency increases the risk of developing cancer [6], and vitamin D supplementation reduces the risk of death from cancer [7]. Vitamin D signaling had a regulatory role in maintaining a healthy immune system, controlling cell proliferation, differentiation and growth and inhibiting angiogenesis [8]. Older adults frequently have serum 25(OH)D insufficiency [9]. The association between 25(OH)D and all-cause mortality in community-based individuals aged > 80 years remains uncertain, with several studies finding significant associations [10,11,12,13] and others reporting negative results [14,15,16,17]. The association between 25(OH)D and all-cause mortality has not been investigated in centenarians.

Two meta-analyses have demonstrated PTH to be a predictor of cardiovascular and all-cause mortality in community-dwelling adults who were middle-aged or older (<80 years) [18,19]. The first meta-analysis of seven studies found that a higher PTH concentration was associated with an increased risk of cardiovascular mortality [18]. The second meta-analysis, which included 10 prospective studies with 31,616 participants, found that an elevated PTH level was associated with a 19% greater risk of all-cause mortality independent of sex and a 68% greater risk of cardiovascular mortality in men [19]. However, the association of PTH with all-cause mortality in centenarians remains unclear. Previous studies have demonstrated significant associations of a higher β-CTX level with myocardial infarction, heart failure, and cardiovascular and all-cause mortality in the general population at higher cardiovascular risk [20,21,22,23]. However, the association of β-CTX with the clinical prognosis in community-dwelling centenarians is largely unknown. Furthermore, it is uncertain whether PTH and β-CTX have any additional effects on the association of 25(OH)D with all-cause mortality in centenarians. Therefore, in this study, we investigated the associations of 25(OH)D, PTH, and β-CTX with all-cause mortality among Chinese community-dwelling centenarians.

## 2. Materials and Methods

### 2.1. Study Design and Population

This prospective cohort population-based study analyzed data from participants in the China Hainan Centenarian Cohort Study (CHCCS), who were recruited between July 2014 and December 2016. The aim of the CHCCS was to establish a multidimensional database of longevity-associated information. The baseline characteristics of this cohort have been reported previously [24,25]. Briefly, 1002 community-dwelling centenarians (180 men and 822 women) were interviewed. Fasting blood samples were collected at the time of enrolment. The most recent survival data were collected between July and September 2021 by the local civil affairs bureau in Hainan. Dates of death for participants who had died were obtained from official reports or death certificates. “Survivors” were considered censored in the latest survey and individuals were considered “lost to follow-up” if they could not be contacted. The present study was approved by the Ethics Committee of the Hainan Hospital of the Chinese People’s Liberation Army General Hospital (No. 301hn11201601). Written informed consent was obtained from all study participants at the time of enrolment.

### 2.2. Covariates

Serum 25(OH)D, PTH, osteocalcin, P1NP, and β-CTX levels were measured by electrochemiluminescence immunoassay using a fully automatic biochemical autoanalyzer (Cobas c702; Roche Products Ltd., Basel, Switzerland) with commercial kits (Roche Diagnostics GmbH, Mannheim, Germany). In accordance with the Endocrine Society guidelines [26], the following serum 25(OH)D thresholds were used: severe deficiency, <10 ng/mL; deficiency and insufficiency, ≥10 ng/mL but <30 ng/mL; and sufficiency, ≥30 ng/mL. The normal reference ranges were used for other variables: serum calcium 2.15–2.55 mmol/L, phosphorus 0.89–1.60 mmol/L, alkaline phosphatase (ALP) 0–130 U/L, P1NP 19–84 μg/L, osteocalcin 11–48 ng/mL, PTH 15–65 pg/mL, and β-CTX 0.55–1.01 ng/mL.

### 2.3. Statistical Analysis

The normality of distribution of continuous variables was determined by the Kolmogorov–Smirnov test. Continuous variables are expressed as the mean and SD if normally distributed and as the median and interquartile range (IQR) if not normally distributed. Categorical variables are shown as the absolute value and percentage. For continuous variables, means were compared between two groups using the unpaired Student’s *t*-test if the variable obeyed a normal distribution or the Mann–Whitney test if not. For categorical variables, groups were compared using the chi-square test. The distribution of time to death is shown by Kaplan–Meier survival curves and compared using the log-rank test.

Both univariate and multivariate Cox regression analyses were performed in this study: the former to estimate the hazard ratio (HR) and 95% CI for mortality and the latter to adjust for confounding factors, including sex, age, body mass index, lifestyle factors (cigarette smoking, alcohol consumption, outdoor activities), comorbidities (incidence of hypertension, diabetes mellitus, cardiovascular disease), past medical history (fractures, surgery), and creatinine, hemoglobin, *C*-reactive protein, homocysteine, and low-density lipoprotein cholesterol levels. Considering the possible nonlinear association of mortality with 25(OH)D, PTH, and β-CTX, these variables were categorized by dividing them into quartiles. The quartile associated with the lowest HR was considered as the reference. Restricted cubic spline (RCS) analyses were performed to analyze the associations of all-cause mortality with 25(OH)D, PTH, and β-CTX as continuous variables in both unadjusted and multivariable adjusted models. All statistical analyses were performed using the IBM SPSS Statistics for Windows, version 19 (IBM Corp., Armonk, NY, USA) and R for Windows, version 4.0.4 (The R Foundation for Statistical Computing, Vienna, Austria). A two-tailed *p*-value of <0.05 was considered statistically significant.

## 3. Results

### 3.1. Baseline Characteristics and Follow-Up

In all, 46 of the 1002 centenarians in the CHCCS had missing values at baseline and 4 were lost during follow-up. The baseline characteristics of the study participants are summarized in Table 1. Of the 952 centenarians who were finally selected, 81.4% were female. The median age was 102 years (IQR, 100, 104). Approximately three-quarters (75.5%) had hypertension, and a minority had diabetes mellitus (9.9%) or a history of fracture (8.6%). The respective percentages of centenarians with abnormal serum calcium, phosphorus, ALP, PTH, P1NP, and osteocalcin levels were 27.1%, 14.8%, 7.2%, 24.3%, 30.5%, and 18.9%. In contrast, 80.8% of participants had an abnormal serum 25(OH)D level, and 75.1% had an abnormal β-CTX level (Figure 1).

During a median follow-up of 32 months (IQR, 15, 55), nearly four-fifths of participants died. The estimated 1-year, 3-year, and 5-year survival rates were 80.0%, 45.7%, and 23.6%, respectively. There was no significant difference in the serum P1NP between centenarians who remained alive and those who died during follow-up. There were statistically significant differences in serum calcium, phosphorus, ALP, and osteocalcin levels between centenarians who remained alive and those who died; however, the differences in serum calcium, phosphorus, ALP, and osteocalcin were relatively small (0.02 mmol/L or a 0.9% decrease, 0.04 mmol/L or a 3.7% increase, 7 U/L or a 9.2% increase, and 2.1 ng/mL or a 7.6% increase, respectively). Compared with centenarians who remained alive, those who died had a significantly lower median 25(OH)D level (21.15 ng/mL (IQR 15.90, 26.80) vs. 24.60 ng/mL (IQR 19.93, 31.33), *p* < 0.001), a significantly higher median PTH level (45.70 pg/mL (IQR 32.51, 63.27) vs. 38.08 pg/mL (IQR 29.90, 49.68), *p* < 0.001), and a significantly higher median β-CTX level (0.42 ng/mL (IQR 0.26, 0.61) vs. 0.34 ng/mL (IQR 0.23, 0.49), *p* < 0.001; Table 1).

### 3.2. Associations of 25(OH)D, PTH, and β-CTX with All-Cause Mortality

The Kaplan–Meier curves, Cox proportional hazards models, and RCS analyses confirmed that serum calcium, phosphorus, ALP, P1NP, and osteocalcin levels were not significantly associated with increased all-cause mortality. However, the Kaplan–Meier survival curves revealed a significant association of a lower 25(OH)D level with increased all-cause mortality (Figure 2A–D). Compared with the highest quartile, the multivariable-adjusted HRs for mortality in the lowest quartile, second quartile, and third quartile were 1.72 (95% CI 1.38–2.14), 1.53 (95% CI 1.24–1.91), and 1.46 (95% CI 1.18–1.81), respectively (Table 2). Mortality was significantly higher in centenarians with a 25(OH)D level below the median (<21.5 ng/mL) than in those with a 25(OH)D level above the median (≥21.5 ng/mL) (1.32, 95% CI 1.13–1.53). Compared with a 25(OH)D level of ≥30 ng/mL, the multivariable-adjusted HRs for mortality in centenarians with 25(OH)D levels of <10 ng/mL and 10–30 ng/mL were 1.86 (95% CI 1.34–2.58) and 1.52 (95% CI 1.25–1.87), respectively. Compared with a 25(OH)D level of ≥30 ng/mL, the multivariable-adjusted HR for mortality in centenarians with 25(OH)D levels of <30 ng/mL (including <10 ng/mL and 10–30 ng/mL) was 1.52 (95% CI 1.24–1.86). The associations of 25(OH)D as a continuous variable with all-cause mortality were further investigated in univariate and multivariate Cox proportional hazard models using RCS analysis (Figure 3). The relationship between 25(OH)D and mortality was linear in both unadjusted and multivariable-adjusted models (*p*_nonlinear_ = 0.518 and *p*_nonlinear_ = 0.526, respectively; Figure 3A,B).

Kaplan–Meier survival curves revealed significant associations of higher PTH levels with increased all-cause mortality (Figure 2E–H). The highest quartile of PTH was associated with a 43% increase in risk of mortality compared with the second quartile (HR 1.43, 95% CI 1.16–1.76); however, the lowest and third quartiles were not associated with a significantly increased risk of mortality compared with the second quartile (Table 2). The mortality rate was significantly higher in centenarians with a PTH level above the median value of >43.89 pg/mL than those with a PTH level below this value (HR 1.22, 95% CI 1.05–1.42). Compared with a PTH level in the range of 15–65 pg/mL, the multivariable-adjusted HRs for mortality in centenarians with a PTH level of <15 pg/mL and >65 pg/mL were 1.09 (95% CI 0.75–1.59) and 1.30 (95% CI 1.08–1.56), respectively. Compared with a PTH level of ≤65 pg/mL, the multivariable-adjusted HR for mortality when PTH was >65 pg/mL was 1.30 (95% CI 1.08–1.56). The relationship between PTH and mortality was approximately J-shaped in both unadjusted and multivariable-adjusted models (*p* _nonlinear_ = 0.001 and *p* _nonlinear_ = 0.010, respectively; Figure 3C,D). For centenarians with a PTH level in the range of 15.56–43.89 pg/mL, the risk of all-cause mortality reached a nadir, with a significant positive association above but a non-significant negative association below this range.

Kaplan–Meier survival curves revealed that a higher β-CTX level was significantly associated with increased all-cause mortality (Figure 2I–L). The highest quartile of β-CTX was associated with a 60% increase in the risk of mortality in comparison with the second quartile (HR 1.60, 95% CI 1.29–1.97). However, the lowest and third quartiles were not associated with a significantly increased risk of mortality in comparison with the second quartile (Table 2). Centenarians with a β-CTX level higher than the median value of ≥0.406 ng/mL had a significantly higher mortality risk than those with a β-CTX level below this value (HR 1.21, 95% CI 1.04–1.40). Compared with a β-CTX level of <0.55 ng/mL, the multivariable-adjusted HRs for mortality at β-CTX levels of 0.55–1.01 ng/mL and >1.01 ng/mL were 1.23 (95% CI 1.04–1.47) and 2.03 (95% CI 1.41–2.92), respectively. Compared with a β-CTX level of <0.55 ng/mL, the multivariable-adjusted HR for mortality at β-CTX levels of ≥0.55 ng/mL (including 0.55–1.01 ng/mL and >1.01 ng/mL) was 1.30 (95% CI 1.10–1.54). The relationship between the β-CTX level and mortality was nearly linear in the unadjusted model but J-shaped in the multivariable-adjusted model (*p* _nonlinear_ = 0.169 and *p* _nonlinear_ = 0.043, respectively; Figure 3E,F). For centenarians with a β-CTX level of 0.227–0.406 ng/mL, the risk of all-cause mortality reached a nadir, with a significant positive association above but a non-significant negative association below this range. It should be noted that centenarians with β-CTX levels in the clinical reference range (0.55–1.01 ng/mL) did have a significantly increased mortality (Figure 3F).

### 3.3. Combined Effects of 25(OH)D, PTH and β-CTX on All-Cause Mortality

According to the median 25(OH)D, PTH, and β-CTX values, we defined the following levels as risk factors: 25(OH)D < 21.5 ng/mL; PTH > 43.89 pg/mL; and β-CTX ≥ 0.406 ng/mL. Centenarians were then categorized into the following four subgroups: G0, no risk factors; G1, one risk factor; G2, two risk factors; and G3, all three risk factors of 25(OH)D < 21.5 ng/mL, PTH > 43.89 pg/mL, and β-CTX ≥ 0.406 ng/mL. Centenarians with one risk factor (G1) did not have a significantly increased risk of all-cause mortality in comparison with those with no risk factors (G0) (HR 1.16, 95% CI 0.93–1.45). In contrast, centenarians with three risk factors (G3) and those with two risk factors (G2) had, respectively, a 2.02-fold and 1.49-fold increase in risk of all-cause mortality in comparison with centenarians with no risk factors (G0) (HR 2.02, 95% CI 1.58–2.59 and HR 1.49, 95% CI 1.19–1.87).

According to the clinical reference values of 25(OH)D, PTH and β-CTX, we defined the following levels as risk factors: 25(OH)D < 30 ng/mL; PTH > 65 pg/mL; and β-CTX ≥ 0.55 ng/mL. Centenarians were categorized into the following four subgroups: G0, no risk factors; G1, one risk factor; G2, two risk factors; and G3, all three risk factors of 25(OH)D < 30 ng/mL, PTH > 65 pg/mL, and β-CTX ≥ 0.55 ng/mL. Centenarians with one risk factor (G1) had only a 42% increase in risk of all-cause mortality compared with those with no risk factors (G0) (HR 1.42, 95% CI 1.11–1.82). In contrast, centenarians with three risk factors (G3) and those with two risk factors (G2) had a respective 2.77-fold and 1.92-fold increase in risk of all-cause mortality in comparison with centenarians without any of these three risk factors (G0) (HR 2.77, 95% CI 1.99–3.85 and HR 1.92, 95% CI 1.46–2.52; Table 3). Thus, not only 25(OH)D but also PTH and β-CTX levels were significantly associated with all-cause mortality. The Kaplan–Meier survival curves for these four subgroups are shown in Figure 4A (categorized by median values) and 4B (categorized by clinical reference values).

## 4. Discussion

The CHCCS included a large cohort of 1002 community-dwelling centenarians with a median follow-up of more than 2.5 years for all participants and of more than 5 years for those who remained alive. The loss to follow-up rate was 0.4%, and nearly 80% of centenarians reached the endpoint of death. Therefore, the CHCCS provides a good model for investigating aging and longevity in community-dwelling centenarians. Our previous work suggested that the serum 25(OH)D level might be associated with activities of daily living, functional dependence, and symptoms of depression at baseline in participants in the CHCCS [27,28,29]. In the present study, we further demonstrated that not only the 25(OH)D level but also the PTH and β-CTX levels were associated with all-cause mortality. Serum P1NP and osteocalcin levels were not significantly associated with mortality.

The association of 25(OH)D with all-cause mortality has been extensively examined in community-based individuals aged > 80 years. Several studies have found a significant association (HR 1.56–2.02) [10,11,12,13], while others have reported a negative association [14,15,16,17]. However, to date, the association between 25(OH)D and all-cause mortality has not been investigated in centenarians. Therefore, the present study provides the first evidence of a linear association between the serum 25(OH)D level and all-cause mortality in centenarians.

The association between PTH and all-cause mortality has rarely been investigated in the general population, especially in the cohort aged >80 years. Furthermore, the available evidence is conflicting. Sambrook et al. showed that PTH was associated with increased mortality in 842 older adults (mean age > 82 years) during a mean follow-up of 31 months [30]. The Helsinki Ageing Study reported a significant impact of elevated serum PTH on all-cause mortality in three age cohorts (75, 80, and 85 years) during a follow-up of 17 years [31]. However, the Malmo Osteoporosis Prospective Risk Assessment cohort, which included 1044 community-dwelling women who were aged >85 years and followed for 10 years, found that PTH was not independently associated with all-cause mortality [32]. However, until now, the association between PTH and all-cause mortality has not been investigated in centenarians. The present study provides the first evidence indicating that the risk of all-cause mortality is significantly increased in centenarians with an elevated PTH but unaffected in those with a PTH level below the normal range. The relationship between PTH and mortality was J-shaped. PTH could affect the calcium level directly or indirectly and has been correlated with vascular calcification and increased cardiovascular risk [33]. Interleukin-1 has been shown to participate in parathyroid cell function and secretion of PTH, and an interleukin-1 receptor antagonist to be able to regulate hypercalcemia [34,35]. Recent studies indicate that interleukin-1 receptor antagonism is a promising therapeutic strategy for cardiovascular disease [36].

The International Osteoporosis Foundation and the International Federation of Clinical Chemistry and Laboratory Medicine recommend P1NP as a marker of bone formation and β-CTX as a marker of bone resorption [2,3]. However, there is limited information on associations of P1NP and β-CTX with mortality. Bager et al. found a U-shaped association of β-CTX with all-cause mortality [37]. A high β-CTX level was also found to be an independent predictor of all-cause mortality in 1112 frail elderly subjects (mean age 86 years) [38]. However, the associations of all-cause mortality with P1NP and β-CTX have not previously been investigated in centenarians. The present study provides the first evidence that a higher β-CTX level (even in the clinical reference range of 0.55–1.01 ng/mL) is associated with a significantly increased risk of all-cause mortality in centenarians. The relationship between β-CTX and mortality was J-shaped. P1NP levels were not associated with all-cause mortality.

Several studies have demonstrated the effect of an interaction between 25(OH)D and PTH on all-cause mortality in the general population but not in centenarians. The Cardiovascular Health Study followed 2312 elderly individuals (aged ≥65 years) for 14 years and showed that all-cause mortality was increased by a further 16% in participants with a 25(OH)D < 15 ng/mL who had a PTH level of ≥65 pg/mL in comparison with their counterparts who had a PTH level of <65 pg/mL (HR 1.43 vs. 1.27) [39]. The CopD Study (a retrospective cohort study conducted in Copenhagen, Denmark) enrolled 34,996 adults (mean age 51 years) who were followed for a median of 3 years. All-cause mortality was significantly higher in participants with 25(OH)D < 25 nmol/L and PTH ≥ 7.6 nmol/L than in those with 25(OH)D < 25 nmol/L and PTH < 7.6 nmol/L (10.3% vs. 4.7%, *p* < 0.0001) [40]. The Health Aging and Body Composition study followed 2638 community-dwelling elderly individuals (aged 71–80 years) for 8.5 years and found that all-cause mortality was a further 60% higher in participants with 25(OH)D < 20 ng/mL if PTH was ≥70 pg/mL than if it was <70 pg/mL (HR 1.96 vs. 1.36) [41]. The Longitudinal Aging Study Amsterdam, which included 1317 elderly individuals (aged 65–85 years) who were followed for 18 years, reported that the relationship of 25(OH)D with all-cause mortality was partly mediated by PTH [42]. However, whether PTH and β-CTX could have an additional effect on the association between 25(OH)D and all-cause mortality in centenarians has not been studied. Our current study provides the first data suggesting that the risk of all-cause mortality is significantly higher in centenarians with at least two risk factors, namely, low 25(OH)D and high PTH and β-CTX, than in those with high 25(OH)D, low PTH and β-CTX.

This study had some limitations. First, we did not analyze the relationship between cause-specific mortality, such as cardiovascular-associated or cancer-associated mortality, because the causes of death were not documented. Second, 25(OH)D, PTH, and β-CTX levels were only measured at baseline. Therefore, we did not analyze any associations between changes in these parameters during follow-up and all-cause mortality.

## 5. Conclusions

To the best of our knowledge, the CHCCS is the largest study with the longest follow-up demonstrating the associations of 25(OH)D, PTH, and β-CTX levels with all-cause mortality in Chinese community-dwelling centenarians. Our findings highlight the importance of PTH and β-CTX levels in the relationship between 25(OH)D insufficiency and all-cause mortality in centenarians.

## Figures and Tables

**Figure 1 nutrients-15-00094-f001:**
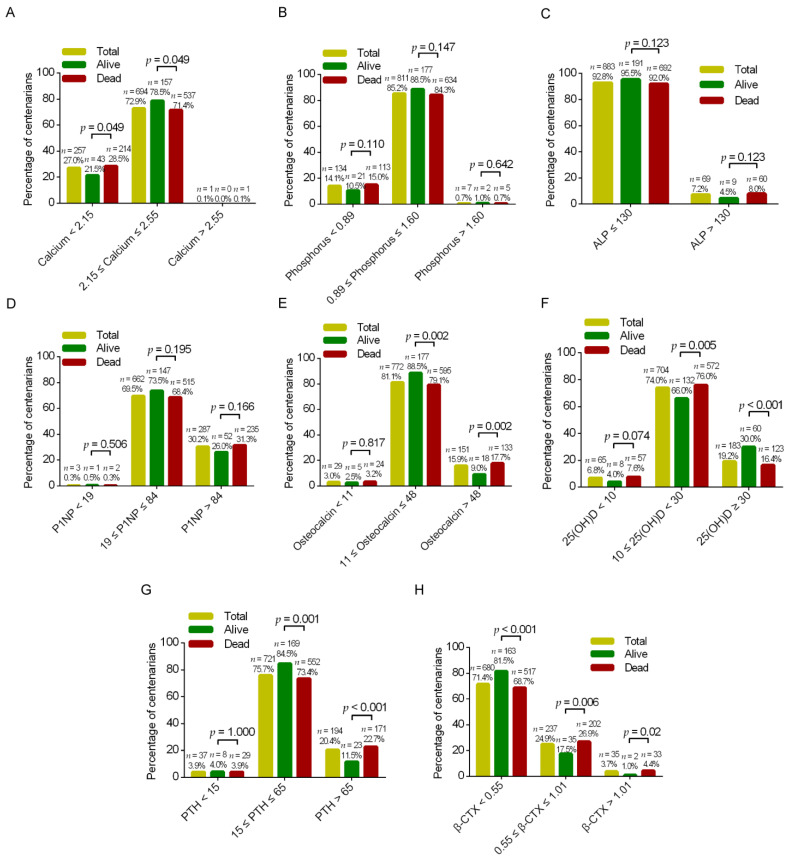
Distributions of centenarians according to serum calcium (mmol/L) (**A**), phosphorus (mmol/L) (**B**), ALP (U/L) (**C**), P1NP (μg/L) (**D**), osteocalcin (ng/mL) (**E**), 25(OH)D (ng/mL) (**F**), PTH (pg/mL) (**G**), and β-CTX (ng/mL) (**H**) (grouped by clinical reference values). 25(OH)D, 25-hydroxyvitamin D; ALP, alkaline phosphatase; β-CTX, β-*C*-terminal telopeptide of type 1 collagen; P1NP, procollagen type 1 *N*-terminal propeptide; PTH, parathyroid hormone.

**Figure 2 nutrients-15-00094-f002:**
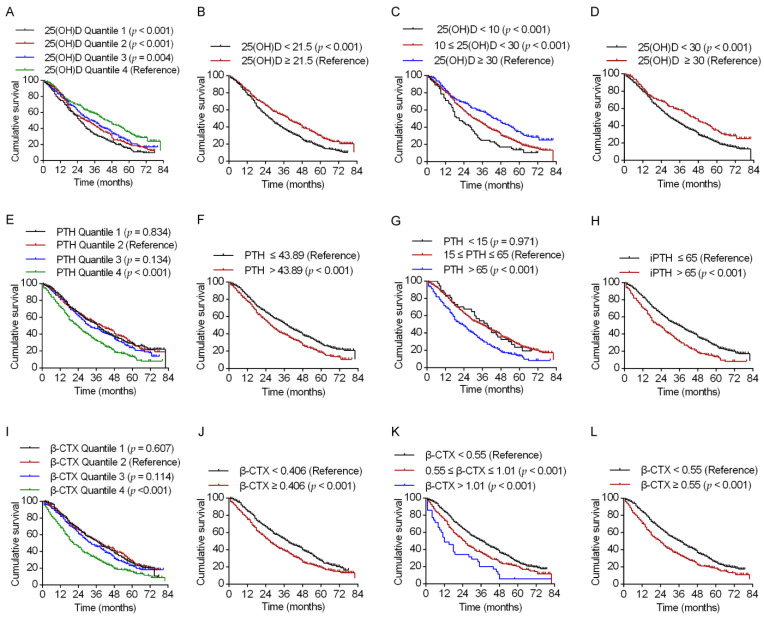
Kaplan–Meier plots and results of the log-rank test for the associations of all-cause mortality with 25(OH)D, PTH, and β-CTX levels. (**A**–**D**) Serum 25(OH)D (ng/mL) by interquartile, median, and clinical reference values. (**E**–**H**) Serum PTH (pg/mL) by interquartile, median, and clinical reference values. (**I**–**L**) Serum β-CTX levels (ng/mL) by interquartile, median, and clinical reference values. 25(OH)D, 25-hydroxyvitamin D; β-CTX, β-*C*-terminal telopeptide of type 1 collagen; PTH, parathyroid hormone.

**Figure 3 nutrients-15-00094-f003:**
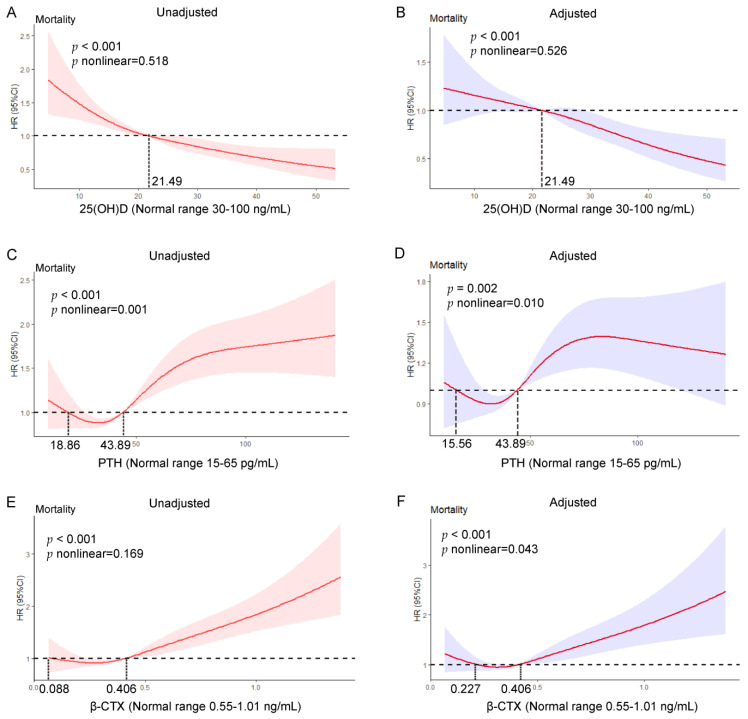
Unadjusted and adjusted associations of serum 25(OH)D, PTH, and β-CTX levels with all-cause mortality in Cox models with restricted cubic spline (RCS) analyses. The solid red line indicates the hazard ratio, and the shaded areas represent the 95% CI. The relationship between 25(OH)D and mortality was linear in both unadjusted (**A**) and multivariable-adjusted models (**B**); The relationship between PTH and mortality was approximately J-shaped in both unadjusted (**C**) and multivariable-adjusted (**D**) models. For centenarians with a PTH level in the range of 15.56–43.89 pg/mL, the risk of all-cause mortality reached a nadir, with a significant positive association above but a non-significant negative association below this range. The relationship between the β-CTX level and mortality was nearly linear in the unadjusted model (**E**) but J-shaped in the multivariable-adjusted model (**F**). For centenarians with a β-CTX level of 0.227–0.406 ng/mL, the risk of all-cause mortality reached a nadir, with a significant positive association above but a non-significant negative association below this range. It should be noted that centenarians with β-CTX levels in the clinical reference range (0.55–1.01 ng/mL) did have a significantly increased mortality. The data are adjusted for demographic factors (age, sex), body mass index, lifestyle factors (cigarette smoking, alcohol consumption, outdoor activities), comorbidities (incidence of hypertension, diabetes mellitus, cardiovascular disease), past medical history (fractures, surgery), and other possible confounding factors (serum creatinine, hemoglobin, *C*-reactive protein, homocysteine, and low-density lipoprotein cholesterol levels). 25(OH)D, 25-hydroxyvitamin D; β-CTX, β-*C*-terminal telopeptide of type 1 collagen; HR, hazard ratio; PTH, parathyroid hormone.

**Figure 4 nutrients-15-00094-f004:**
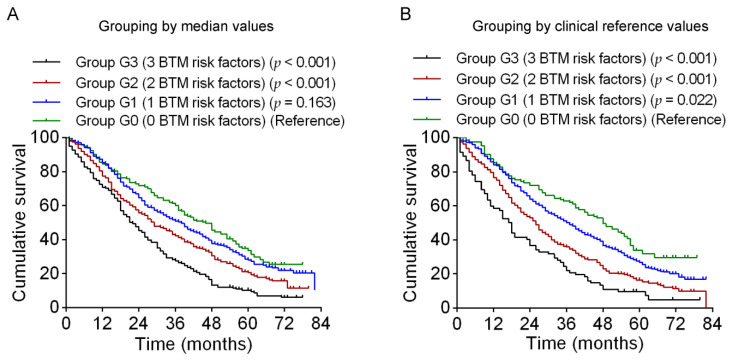
Kaplan–Meier plots and results of log-rank tests for the associations of all-cause mortality with the combined risk factors of lower 25(OH)D, higher PTH, and higher β-CTX. Centenarians were categorized into the four following subgroups: G0, no risk factors; G1, one risk factor; G2, two risk factors; G3, all three risk factors of (**A**) 25(OH)D < 21.5 ng/mL, PTH > 43.89 pg/mL, and β-CTX ≥ 0.406 ng/mL or (**B**) 25(OH)D < 30 ng/mL, PTH > 65 pg/mL, and β-CTX ≥ 0.55 ng/mL. BTM, bone turnover markers; PTH, parathyroid hormone.

**Table 1 nutrients-15-00094-t001:** Baseline demographic and clinical characteristics of the entire cohort of centenarians.

Characteristics	Total	Alive Throughout the Study	Dead during the Study	*p* Value
Participants (%)	952 (100.0)	200 (21.1)	752 (78.9)	-
Follow-up time (months) Median (IQR)	32 (15, 55)	61 (56, 71)	24 (12, 41)	<0.001
Age (Years) Median (IQR)	102 (100, 104)	101 (100, 104)	102 (100, 104)	0.552
Female (%)	775 (81.4)	167 (83.5)	608 (80.9)	0.392
Hypertension (%)	719 (75.5)	156 (78.0)	563 (78.9)	0.360
Diabetes mellitus (%)	94 (9.9)	18 (9.0)	76 (10.1)	0.641
Previous fractures (%)	82 (8.6)	13 (6.5)	69 (9.2)	0.231
Calcium (mmol/L) (Mean ± SD) (Normal range 2.15–2.55)	2.21 ± 0.11	2.23 ± 0.11	2.21 ± 0.11	0.005
Phosphorus (mmol/L) (Mean ± SD) (Normal range 0.89–1.6)	1.06 ± 0.17	1.09 ± 0.18	1.05 ± 0.16	0.003
ALP (U/L) Median (IQR) (Normal range 0–130)	81 (67, 102)	76 (64, 96)	83 (67, 103)	0.010
P1NP (μg/L) Median (IQR) (Normal range 19–84)	65.03 (47.00, 91.00)	61.50 (44.25, 85.00)	66.00 (48.00, 92.75)	0.090
Osteocalcin (ng/mL) Median (IQR) (Normal range 11–48)	29.28 (20.53, 40.67)	27.58 (19.65, 36.37)	29.68 (20.79, 41.77)	0.010
25(OH)D (ng/mL) Median (IQR) (Normal range 30–100)	21.60 (16.50, 28.10)	24.60 (19.93, 31.33)	21.15 (15.90, 26.80)	<0.001
PTH (pg/mL) Median (IQR) (Normal range 15–65)	43.91 (31.81, 60.76)	38.08 (29.90, 49.68)	45.70 (32.51, 63.27)	<0.001
β-CTX (ng/mL) Median (IQR) (Normal range 0.55–1.01)	0.40 (0.25, 0.58)	0.34 (0.23, 0.49)	0.42 (0.26, 0.61)	<0.001

Abbreviations: 25(OH)D, 25-hydroxyvitamin D; ALP, alkaline phosphatase; β-CTX, β-*C*-terminal telopeptide of type 1 collagen; IQR, interquartile range; P1NP, procollagen type 1 *N*-terminal propeptide; PTH, parathyroid hormone; SD, standard deviation.

**Table 2 nutrients-15-00094-t002:** Cox regression analyses of the associations of 25(OH)D, PTH, and β-CTX with all-cause mortality in centenarians.

Variables	Grouped by Interquartile Values				Grouped by Median Values		Grouped by Clinical Values (Three Groups)			Grouped by Clinical Values (Two Groups)	
25(OH)D (ng/mL)	Quantile 1 (<16.6, N = 239)	Quantile 2 (16.6–21.5, N = 225)	Quantile 3 (21.5–28.1, N = 253)	Quantile 4 (>28.1, N = 235)	<21.5 (N = 458)	≥21.5 (N = 494)	<10 (N = 65)	10–30 (N = 704)	≥30 (N = 183)	<30 (N = 769)	≥30 (N = 183)
Unadjusted	1.84 (1.49, 2.26)	1.51 (1.22, 1.86)	1.35 (1.10, 1.67)	1.00 (Ref)	1.42 (1.23, 1.64)	1.00 (Ref)	2.05 (1.50, 2.81)	1.44 (1.18, 1.75)	1.00 (Ref)	1.48 (1.22, 1.79)	1.00 (Ref)
*p* values	<0.001	<0.001	0.005	-	<0.001	-	<0.001	<0.001	-	<0.001	-
Adjusted	1.72 (1.38, 2.14)	1.53 (1.24, 1.91)	1.46 (1.18, 1.81)	1.00 (Ref)	1.32 (1.13, 1.53)	1.00 (Ref)	1.86 (1.34, 2.58)	1.52 (1.25, 1.87)	1.00 (Ref)	1.52 (1.24, 1.86)	1.00 (Ref)
*p* values	<0.001	<0.001	0.001	-	<0.001	-	<0.001	<0.001	-	<0.001	-
PTH (pg/mL)	Quantile 1 (<31.82, N = 238)	Quantile 2 (31.82–43.89, N = 238)	Quantile 3 (43.89–60.68, N = 238)	Quantile 4 (>60.68, N = 238)	≤43.89 (N = 476)	>43.89 (N = 476)	<15 (N = 37)	15–65 (N = 721)	>65 (N = 194)	≤65 (N = 758)	>65 (N = 194)
Unadjusted	1.02 (0.83, 1.26)	1.00 (Ref)	1.17 (0.95, 1.44)	1.71 (1.40, 2.09)	1.00 (Ref)	1.38 (1.20, 1.60)	1.01 (0.69, 1.46)	1.00 (Ref)	1.61 (1.36, 1.92)	1.00 (Ref)	1.61 (1.36, 1.91)
*p* values	0.831	-	0.136	<0.001	-	<0.001	0.979	-	<0.001	-	<0.001
Adjusted	1.09 (0.88, 1.35)	1.00 (Ref)	1.15 (0.94, 1.42)	1.43 (1.16, 1.76)	1.00 (Reference)	1.22 (1.05, 1.42)	1.09 (0.75, 1.59)	1.00 (Ref)	1.30 (1.08, 1.56)	1.00 (Ref)	1.30 (1.08, 1.56)
*p* values	0.417	-	0.182	0.001	-	0.011	0.643	-	0.005	-	0.005
β-CTX (ng/mL) (Interquartile)	Quantile 1 (<0.248, N = 240)	Quantile 2 (0.248–0.406, N = 237)	Quantile 3 (0.406–0.579, N = 238)	Quantile 4 (>0.579, N = 237)	<0.406 (N = 477)	≥0.406 (N = 475)	<0.55 (N = 680)	0.55–1.01 (N = 237)	>1.01 (N = 35)	<0.55 (N = 680)	≥0.55 (N = 272)
Unadjusted	1.05 (0.85, 1.29)	1.00 (Ref)	1.17 (0.95, 1.44)	1.70 (1.39, 2.08)	1.00 (Ref)	1.37 (1.19, 1.58)	1.00 (Ref)	1.36 (1.15, 1.60)	2.30 (1.61, 3.28)	1.00 (Ref)	1.44 (1.23, 1.68)
*p* values	0.670	-	0.129	<0.001	-	<0.001	-	<0.001	<0.001	-	<0.001
Adjusted	1.17 (0.95, 1.45)	1.00 (Ref)	1.11 (0.90, 1.36)	1.60 (1.29, 1.97)	1.00 (Ref)	1.21 (1.04, 1.40)	1.00 (Ref)	1.23 (1.04, 1.47)	2.03 (1.41, 2.92)	1.00 (Ref)	1.30 (1.10, 1.54)
*p* values	0.137	-	0.341	<0.001	-	0.016	-	0.018	<0.001	-	0.002

The data are adjusted for demographic factors (age, sex), body mass index, lifestyle factors (cigarette smoking, alcohol consumption, outdoor activities), comorbidities (incidence of hypertension, diabetes mellitus, cardiovascular disease), past medical history (fractures, surgery), markers of bone metabolism (serum calcium, phosphorus, alkaline phosphatase, procollagen type 1 *N*-terminal propeptide, osteocalcin, PTH, 25(OH)D, and β-CTX), and other possible confounding factors (serum creatinine, hemoglobin, *C*-reactive protein, homocysteine, and low-density lipoprotein cholesterol levels). Abbreviations: 25(OH)D, 25-hydroxyvitamin D; β-CTX, β-*C*-terminal telopeptide of type 1 collagen; PTH, parathyroid hormone; Ref, reference.

**Table 3 nutrients-15-00094-t003:** Cox regression analyses for the associations of all-cause mortality with the combined risk factors of low 25(OH)D, high PTH and β-CTX levels in centenarians.

Groups		Sample Size (N)	25(OH)D (ng/mL)	PTH (pg/mL)	β-CTX (ng/mL)	Unadjusted HR, 95% CI and *p* Values	Adjusted HR, 95% CI and *p* Values
Grouped by median values	G3 (3 risk factors)	174	<21.5	>43.89	≥0.406	2.20 (1.73, 2.79) (*p* < 0.001)	2.02 (1.58, 2.59) (*p* < 0.001)
	G2 (2 risk factors)	282	<21.5	>43.89	<0.406	1.51 (1.21, 1.88) (*p* < 0.001)	1.49 (1.19, 1.87) (*p* = 0.001)
				≤43.89	≥0.406		
			≥21.5	>43.89	≥0.406		
	G1 (1 risk factor)	323	≥21.5	>43.89	<0.406	1.17 (0.94, 1.46) (*p =* 0.171)	1.16 (0.93, 1.45) (*p* = 0.196)
			≥21.5	≤43.89	≥0.406		
			<21.5	≤43.89	<0.406		
	G0 (0 risk factor)	173	≥21.5	≤43.89	<0.406	1.00 (Reference)	1.00 (Reference)
Grouped by clinical reference values	G3 (3 risk factors)	82	<30	>65	≥0.55	2.82 (2.05, 3.87) (*p* < 0.001)	2.77 (1.99, 3.85) (*p* < 0.001)
	G2 (2 risk factors)	240	<30	>65	<0.55	1.89 (1.45, 2.45) (*p* < 0.001)	1.92 (1.46, 2.52) (*p* < 0.001)
				≤65	≥0.55		
			≥30	>65	≥0.55		
	G1 (1 risk factor)	509	≥30	>65	<0.55	1.32 (1.03, 1.68) (*p =* 0.026)	1.42 (1.11, 1.82) (*p* = 0.006)
			≥30	≤65	≥0.55		
			<30	≤65	<0.55		
	G0 (0 risk factor)	121	≥30	≤65	<0.55	1.00 (Reference)	1.00 (Reference)

Risk factors included (1) decreased 25(OH)D, (2) increased PTH, and (3) increased β-CTX levels. The data are adjusted for demographic factors (age, sex), body mass index, lifestyle factors (cigarette smoking, alcohol consumption, outdoor activities), comorbidities (incidence of hypertension, diabetes mellitus, cardiovascular disease), past medical history (fractures, surgery), markers of bone metabolism (serum calcium, phosphorus, ALP, P1NP, osteocalcin, PTH, 25(OH)D and β-CTX), and other possible confounding factors (serum creatinine, hemoglobin, *C*-reactive protein, homocysteine and low-density lipoprotein cholesterol levels). Abbreviations: 25(OH)D, 25-hydroxyvitamin D; β-CTX, β-*C*-terminal telopeptide of type 1 collagen; HR, hazard ratio; PTH, parathyroid hormone.

## Data Availability

The data that support the findings of this study are available from the corresponding author upon reasonable request.

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
