# Peer review of "Associations of Serum 25(OH)D, PTH, and β-CTX Levels with All-Cause Mortality in Chinese Community-Dwelling Centenarians"

_nutrients, 2022, doi:10.3390/nu15010094_

Round 1

Reviewer 1 Report

Manuscript titled "Associations of serum 25(OH)D, PTH, and β-CTX levels with all-cause mortality in Chinese community-dwelling centenarians" is a clear and original article describing associations of serum 25(OH)D, PTH, and β-CTX levels with all-cause mortality. The overall structure is of good quality, methods are clear and results are easy to read. However, I suggest to the authors to improve the manuscript in some parts:

1. A more appropriate description of vitamin d multiple beneficial effects against cancer and cardiovascular diseases should be done, with a proper description of pathways involved .

2. A more appropriate description, in discussion, of other metabolic thyroid-and vitamin d related cardiovascular risk factors in these patients; i.e, how PTH levels should be associated to high interleukin 1 levels and how selective antagonists of IL1 should be promising therapeutic drugs. 

Author Response

Response to Reviewer #1’s comments

  1. A more appropriate description of vitamin d multiple beneficial effects against cancer and cardiovascular diseases should be done, with a proper description of pathways involved.

Response: Thanks for your comments. In the revised manuscript, we have provided with more evidence of the multiple beneficial effects of vitamin D on cancer and cardiovascular diseases, with the description of the pathways involved in the protection effects in the introduction section.

“Vitamin D has been shown to have a protective effect on cardiovascular risk in both observational studies and randomized clinical trials [4]. Vitamin D might have favorable interactions with the cardiovascular system, including inhibition of the renin-angiotensin-aldosterone pathway, insulin sensitization, immune regulation, and antifibrotic, antihypertrophic, and antiatherosclerotic effects [5]. Vitamin D deficiency increases the risk of developing cancer [6], and vitamin D supplementation reduces the risk of death from cancer [7]. Vitamin D signaling had a regulatory role in maintaining a healthy immune system, controlling cell proliferation, differentiation and growth and inhibiting angiogenesis [8].”

  1. A more appropriate description, in discussion, of other metabolic thyroid-and vitamin d related cardiovascular risk factors in these patients; i.e, how PTH levels should be associated to high interleukin 1 levels and how selective antagonists of IL1 should be promising therapeutic drugs. 

Response: Thanks for your suggestions. In the revised manuscript, we have provided with the relationship between PTH and cardiovascular diseases, with the possible role of interleukin 1 in it. Meanwhile, the role of interleukin-1 receptor antagonist as a potential therapeutic strategy for cardiovascular diseases was also discussed in the discussion section.

“PTH could affect the calcium level directly or indirectly and has been correlated with vascular calcification and increased cardiovascular risk [33]. Interleukin-1 has been shown to participate in parathyroid cell function and secretion of PTH, and an interleukin-1 receptor antagonist to be able to regulate hypercalcemia [34,35]. Recent studies indicate that interleukin-1 receptor antagonism is a promising therapeutic strategy for cardiovascular disease [36].”

References

  1. Michos, E.D.; Cainzos-Achirica, M.; Heravi, A.S.; Appel, L.J. Vitamin D, Calcium Supplements, and Implications for Cardiovascular Health: JACC Focus Seminar. Journal of the American College of Cardiology 2021, 77, 437-449, doi:10.1016/j.jacc.2020.09.617.
  2. Zittermann, A.; Trummer, C.; Theiler-Schwetz, V.; Lerchbaum, E.; März, W.; Pilz, S. Vitamin D and Cardiovascular Disease: An Updated Narrative Review. International journal of molecular sciences 2021, 22, doi:10.3390/ijms22062896.
  3. Feldman, D.; Krishnan, A.V.; Swami, S.; Giovannucci, E.; Feldman, B.J. The role of vitamin D in reducing cancer risk and progression. Nature reviews. Cancer 2014, 14, 342-357, doi:10.1038/nrc3691.
  4. Zhang, Y.; Fang, F.; Tang, J.; Jia, L.; Feng, Y.; Xu, P.; Faramand, A. Association between vitamin D supplementation and mortality: systematic review and meta-analysis. BMJ (Clinical research ed.) 2019, 366, l4673, doi:10.1136/bmj.l4673.
  5. El-Sharkawy, A.; Malki, A. Vitamin D Signaling in Inflammation and Cancer: Molecular Mechanisms and Therapeutic Implications. Molecules (Basel, Switzerland) 2020, 25, doi:10.3390/molecules25143219.
  6. Bollerslev, J.; Sjöstedt, E.; Rejnmark, L. Cardiovascular consequences of parathyroid disorders in adults. Annales d'endocrinologie 2021, 82, 151-157, doi:10.1016/j.ando.2020.02.003.
  7. Toribio, R.E.; Kohn, C.W.; Capen, C.C.; Rosol, T.J. Parathyroid hormone (PTH) secretion, PTH mRNA and calcium-sensing receptor mRNA expression in equine parathyroid cells, and effects of interleukin (IL)-1, IL-6, and tumor necrosis factor-alpha on equine parathyroid cell function. Journal of molecular endocrinology 2003, 31, 609-620, doi:10.1677/jme.0.0310609.
  8. Guise, T.A.; Garrett, I.R.; Bonewald, L.F.; Mundy, G.R. Interleukin-1 receptor antagonist inhibits the hypercalcemia mediated by interleukin-1. Journal of bone and mineral research : the official journal of the American Society for Bone and Mineral Research 1993, 8, 583-587, doi:10.1002/jbmr.5650080509.
  9. Buckley, L.F.; Abbate, A. Interleukin-1 blockade in cardiovascular diseases: a clinical update. European heart journal 2018, 39, 2063-2069, doi:10.1093/eurheartj/ehy128.

Reviewer 2 Report

In China, the average life expectancy has recently increased to nearly 80 years, and more than 50,000 people now live longer than 100 years. Most of them are women.

This paper is an analysis based on interviews and blood samples from the China Hainan Centenarian Cohort Study (CHCCS), a cohort database related to longevity in China.

The results are very informative and have not been revised.

Author Response

Response to Reviewer #2’s comments

  1. In China, the average life expectancy has recently increased to nearly 80 years, and more than 50,000 people now live longer than 100 years. Most of them are women.

This paper is an analysis based on interviews and blood samples from the China Hainan Centenarian Cohort Study (CHCCS), a cohort database related to longevity in China.

The results are very informative and have not been revised.

Response: Thanks for your comments. In the revised manuscript, we have further revised the manuscript according to these constructive comments, with graphic abstract provided and the English grammar edited.